Microbial metagenome-assembled genomes of the Fram Strait from short and long read sequencing platforms

Priest Taylor 1
Orellana Luis H. lorellan@mpi-bremen.de 1
Huettel Bruno 2
Fuchs Bernhard M. 1
Amann Rudolf 1
1 Department of Molecular Ecology, Max Planck Institute for Marine Microbiology , Bremen , Germany
2 Max-Planck-Genome-Centre Cologne , Cologne , Germany
Makhalanyane Thulani
Electronic publication date: 2021 Jun 30
Publication date: 2021
Volume: 9
Electronic Location ID: e11721
Received 2021 Apr 8; Accepted 2021 Jun 14
Copyright: ©2021 Priest et al.
Copyright year: 2021
Copyright holder: Priest et al.
License: This is an open access article distributed under the terms of the Creative Commons Attribution License, which permits unrestricted use, distribution, reproduction and adaptation in any medium and for any purpose provided that it is properly attributed. For attribution, the original author(s), title, publication source (PeerJ) and either DOI or URL of the article must be cited.
License URL: https://creativecommons.org/licenses/by/4.0/

Keywords: Arctic, Microbiology, Metagenomics, Metagenome-assembled genomes, Microbial ecology

Funding: The Max Planck Society This work was funded by the Max Planck Society. The funders had no role in study design, data collection and analysis, decision to publish, or preparation of the manuscript.

==============================
The impacts of climate change on the Arctic Ocean are manifesting throughout the ecosystem at an unprecedented rate. Of global importance are the impacts on heat and freshwater exchange between the Arctic and North Atlantic Oceans. An expanding Atlantic influence in the Arctic has accelerated sea-ice decline, weakened water column stability and supported the northward shift of temperate species. The only deep-water gateway connecting the Arctic and North Atlantic and thus, fundamental for these exchange processes is the Fram Strait. Previous research in this region is extensive, however, data on the ecology of microbial communities is limited, reflecting the wider bias towards temperate and tropical latitudes. Therefore, we present 14 metagenomes, 11 short-read from Illumina and three long-read from PacBio Sequel II, of the 0.2–3 µm fraction to help alleviate such biases and support future analyses on changing ecological patterns. Additionally, we provide 136 species-representative, manually refined metagenome-assembled genomes which can be used for comparative genomics analyses and addressing questions regarding functionality or distribution of taxa.

Introduction

The Arctic Ocean is a critical component in the maintenance of Earth’s energy balance and the regulation of global climate. Of major importance is the exchange of heat and freshwater between the Arctic and North Atlantic Oceans. The northward transport of Atlantic water is the primary source of heat to the interior of the Arctic Ocean and is vital for water column stability (Rudels et al., 1994; Spall, 2013). Similarly, the southward transport of Arctic water plays a fundamental role in the thermohaline circulation through the formation of North Atlantic Deepwater (McGuire et al., 2006). As is becoming increasingly evident, these processes are experiencing pronounced perturbations as a result of climate change. The inflowing Atlantic water has doubled in volume in the last 30 years (Oziel et al., 2020) and increased in temperature by 1.4 °C (Neukermans, Oziel & Babin, 2018). This has manifested in an expanding Atlantic presence across the Eurasian Arctic (Polyakov et al., 2017), contributing to sea-ice decline (Lind, Ingvaldsen & Furevik, 2018), warmer subsurface temperatures, a weakening of water column stability and the northward expansion of temperate organisms (Neukermans, Oziel & Babin, 2018; Oziel et al., 2020). These phenomena are evidence of a much broader and more long-term transition in the state of the Arctic Ocean ecosystem.

The primary region of exchange between the Arctic and North Atlantic Oceans is the Fram Strait. This 450 km wide deep-water gateway, situated between Greenland and Svalbard, is a convergence zone of two distinct hydrographic regimes. The West Spitsbergen Current transports warm and salty Atlantic water northward through the eastern Fram Strait whilst in the western Fram Strait, the East Greenland Current is responsible for 51% of the Arctic Ocean freshwater export (Serreze et al., 2006). The convergence of these opposing currents, in a relatively narrow geographical area, provides an invaluable opportunity to investigate the ongoing impacts of climate change on the Arctic Ocean.

Despite decades of research in this region, information regarding the ecology of microbial communities is limited. As primary production increases in the Arctic (Lewis, Dijken & Arrigo, 2020) and coastal influences such as thawing permafrost become more pronounced (Lantuit & Pollard, 2008; Vonk et al., 2012), the availability, quantity and composition of organic matter will change substantially. The primary degraders of organic matter in the marine environment are heterotrophic microbes (Azam, 1998). Therefore, characterising their ecology and, in particular, their functional capabilities, may provide insights as to how the Arctic Ocean ecosystem will cope with, and adapt to changing conditions. Currently, a powerful method for addressing such topics is metagenomics.

Widespread efforts in environmental sequencing and the retrieval of metagenome-assembled genomes (MAGs) have been largely directed towards epipelagic communities in temperate and tropical latitudes whilst the polar regions are far less studied. Therefore, we aim to contribute to the alleviation of these biases and present 14 metagenomes (11 from Illumina HiSeq 3000 and three from PacBio Sequel II sequencing platforms) of the 0.2–3 µm fraction from the Fram Strait region. These metagenomic datasets were assembled, binned and manually curated to generate 136 species-representative MAGs.

Materials & Methods

Sample collection and read generation

Samples (n = 11) were collected between July–August of 2018 from the Fram Strait region whilst onboard the RV Polarstern (PS114 cruise). A map of the sampling locations was generated using publically available bathymetric data (GEBCO Compilation Group, 2020; Jakobsson et al., 2020) and edited using QGIS v3.14.16-Pi (QGIS.org, 2021) (Fig. 1). The samples were mostly derived from the deep chlorophyll maximum layer, determined using the in-built Fluorimeter of the Conductivity, Temperature and Depth (CTD)-Rosette sampler during the downcast, and detailed information on their location is provided in Table S1. For each sample, 1 L of seawater was retrieved with a CTD-Rosette sampler and sequentially filtered through a 10 µm, 3 µm and 0.2 µm polycarbonate membrane filter (47 mm diameter) for size fractionation. The full 1 L was filtered through the 10 µm fraction whereas the sample was divided into 2 × 500 ml for the smaller size fractions. The filters were immediately frozen and kept at −80 °C until the extraction of DNA. DNA was extracted from one of the 0.2–3 µm fraction filters for each sample (500 ml filtered) following a modified SDS-based extraction method after Zhou, Bruns & Tiedje (1996). The quality of extraction and quantification of DNA was determined using a Qubit 2.0 Fluorometer (Invitrogen, Darmstadt, Germany) (Table 1).

Figure 1 Bathymetric map of sampling locations.

The visualised stations were sampled during the PS114 Polarstern cruise in July and August of 2018. The colour scale from red to green indicates depth changes that are more accurately represented by the contour lines. Stations were categorised into three water masses based on temperature and salinity measurements. Squares represent Arctic water mass, pentagons represent Atlantic water mass and circles represent mixed water mass.

Table 1 Summary statistics on raw and assembled metagenomes.

Water mass abbreviations are ‘Arc = Arctic’, ‘Mix = mixed’, ‘Atl = Atlantic’. Library names containing ‘4514’ or ‘4502’ are derived from Illumina HiSeq3000 reads that were assembled with Megahit whilst those containing ‘4571’ are derived from PacBio Sequel II reads that were assembled with MetaFlye.

Sample name	Water mass	DNA yield (ng)	Sequencing platform	Raw read number	Average sequence length (bp)	Estimated coverage	Total assembly length (Mbp)	L50	N50	Number of contigs	Max contig length (Mbp)	
FRAM18_4514_A	Arc	25.1	Illumina HiSeq 3000	84,692,914	150	0.79	1,103.249	561	409611	2087649	0.196	
FRAM18_4514_B	Mix	530.8	Illumina HiSeq 3000	102,164,804	150	0.86	1,015.001	799	221669	1644515	0.263	
FRAM18_4514_C	Arc	65.4	Illumina HiSeq 3000	73,810,275	150	0.72	1,187.025	654	370621	2109170	0.302	
FRAM18_4514_D	Atl	613.7	Illumina HiSeq 3000	81,643,335	150	0.73	1,290.853	690	365403	2237087	0.230	
FRAM18_4502_E	Arc	243.2	Illumina HiSeq 3000	106,969,873	150	0.8	1,645.447	676	492678	2780604	0.668	
FRAM18_4502_F	Arc	231.6	Illumina HiSeq 3000	81,593,071	150	0.74	1,065.963	628	373268	1860335	0.249	
FRAM18_4502_G	Arc	189.9	Illumina HiSeq 3000	87,777,577	150	0.76	1,481.554	635	519881	2609315	0.382	
FRAM18_4502_H	Arc	277.9	Illumina HiSeq 3000	97,201,780	150	0.82	1,314.445	632	449089	2337881	0.435	
FRAM18_4571_H	Arc	Pacbio Sequel II	625,530	9653	0.85	386.515	61541	1295	9700	1.594	
FRAM18_4502_I	Mix	303	Illumina HiSeq 3000	71,890,475	150	0.8	961.129	758	231982	1292104	0.409	
FRAM18_4571_I	Mix	Pacbio Sequel II	455,246	9665	0.88	317.028	71964	904	7545	2.067	
FRAM18_4502_J	Mix	359	Illumina HiSeq 3000	117,925,127	150	0.86	914.466	856	195028	1526032	0.408	
FRAM18_4502_K	Atl	190.7	Illumina HiSeq 3000	107,105,259	150	0.83	1,658.809	722	459518	2090439	0.396	
FRAM18_4571_K	Atl	Pacbio Sequel II	552,543	8599	0.84	420.259	67691	1226	9944	2.309	
Notes.

Water mass abbreviations are ‘Arc, Arctic’; ‘Mix, mixed; ‘Atl, Atlantic;’. Library names containing ’4514’ or ’4502’ are derived from Illumina HiSeq3000 reads that were assembled with Megahit whilst those containing ’4571’ are derived from PacBio Sequel II reads that were assembled with MetaFlye.

All 11 samples were sequenced at the Max Planck Genome Centre in Cologne, Germany. Sequencing was performed on an Illumina HiSeq 3000 platform, following an ultra-low input library preparation protocol. This resulted in 71–118 million paired-end reads per sample of 150 bp in length (Table 1). Additionally, three of the samples were sequenced on a PacBio Sequel II platform, following an ultra-low library preparation protocol. The ultra-low PacBio library protocol involves a long-range PCR step for AT- and GC-rich sequences followed by a size selection step (removal of sequences < 4.5 kbp). The three samples were barcoded, pooled into a single library and sequenced on a single SMRT Cell. The circular consensus sequencing method was used, generating HiFi reads with >99% per-base accuracy and an output of 4–6 Gbp per sample with an average read length of 8.6–9.6 kbp (Table 1). All sequencing runs were performed without positive or negative controls. To provide an insight into the quality of the obtained sequences, plots of the base quality scores across reads produced by FastqQC (Illumina metagenomes) and NanoPlot v1.32.1 (De Coster et al., 2018) (PacBio metagenomes) are provided for a selection of the metagenomes (Fig. S1).

Metagenomic assembly and binning of Illumina reads

Prior to read analysis, Nonpareil v3.3.3 (Rodriguez & Konstantinidis, 2014; Rodriguez et al., 2018) was used to provide an estimate of the level of coverage of each metagenome (Table 1). As Nonpareil was designed for short reads only, the long PacBio reads were sheared into 150 bp fragments to allow for comparative analysis to the Illumina reads. Coverage values ranged from 0.72–0.88, indicating a high level of coverage with the chosen sequencing depth. Low quality reads and adapters were removed from the Illumina dataset using BBDuk of the BBtools package v38.73 (http://bbtools.jgi.doe.gov/) (parameters: ktrim = r, k = 29, mink = 12, hdist = 1, tbo = t, tpe = t, qtrim = rl, trimq = 20, minlength = 100). Megahit v1.2.9 (Li et al., 2016) (parameters: –presets meta-large, –cleaning-rounds 5) was used to assemble short-read metagenomes individually (Table 1).

Quality trimmed reads were mapped to the assemblies using BBMap of the BBtools package (Bushnell, 2014) (parameters: minid = 99, idfilter = 97) to provide coverage information for binning. The recovery of MAGs was then performed in a multi-step approach. Firstly, contigs >2.5 kbp in length were binned using three different programs with default settings: Metabat2 v2.12.1 (Kang et al., 2019), Concoct v1.1.0 (Alneberg et al., 2014) and MaxBin2 v2.2.7 (Wu, Simmons & Singer, 2016). A consensus set of non-redundant bins was subsequently retrieved using DasTool v1.1.1 (Sieber et al., 2018) and taxonomically classified using CheckM v1.1.2 (Parks et al., 2015). In the second step, bins assigned to same taxonomic class were concatenated into a single file and used to recruit raw reads with BBmap (minid = 95, idfilter = 95). The successfully mapped reads from each class were assembled using Megahit (parameters: –presets meta-sensitive –cleaning-rounds 5). The binning pipeline described above was then repeated for each class-level assembly. The taxonomic reassembly was performed as it can greatly improve the quality of MAGs produced through increased contiguity and reduced contamination. The completeness and contamination were determined using CheckM and those that were < 50% complete were removed from further analysis. Reads were recruited to the remaining 218 MAGs using BBmap (parameters: minid = 99 idfilter = 97) to generate coverage information. The MAGs and resulting sequence alignment map files were processed with the metagenomics pipeline implemented in Anvi’o v6.1 (Eren et al., 2015). All 218 MAGs were then manually refined using the anvio interactive interface with the anvi-refine function to inspect coverage and reduce contamination where necessary.

Metagenomic assembly and binning of PacBio reads

PacBio HiFi reads were subject to error correction using the program FMLRC v1.0 (Wang et al., 2018) with the Illumina quality trimmed reads as a reference. The reads were processed in a similar pipeline as described for the Illumina reads except, metaFlye v2.6 (Kolmogorov et al., 2020) (parameters: –pacbio-hifi, -i 5, –genome-size 150 m) was used for assembly (Table 1). To obtain coverage information for binning, the Illumina reads derived from the same sample were used for mapping, using the same parameters described above, due to the robustness of short-read mapping tools. Taxonomic reassembly was not performed for the PacBio dataset due to the high quality of generated MAGs from single metagenome assemblies. The three PacBio assemblies resulted in 128 consensus MAGs being retrieved after removing those with <50% completeness. All MAGs were then manually refined as described for the Illumina dataset.

The 346 manually refined MAGs were compared using FastANI v1.9 (Jain et al., 2018) and grouped into species-level clusters with a genome alignment threshold of 30% and a 95% identity threshold. The highest quality MAG from each cluster (determined by completeness, contamination, N50, number of contigs and the presence of rRNA genes) was designated the representative (Table S2) and used for phylogenetic tree reconstruction. Quality trimmed reads from the Illumina metagenomes were recruited to each species-representative MAG and relative abundances calculated (Table S3). A schematic diagram is provided that summarises the workflow used to analyse the metagenomic data (Fig. 2).

Figure 2 Schematic diagram of the workflow used to process the metagenomic data.

Phylogenetic assessment of MAGs

The taxonomic classification of MAGs was performed using two approaches. Firstly, the classify_wf pipeline of GTDB-tk v1.0.2 (Chaumeil et al., 2020; Parks et al., 2020) (Release89) was used. Secondly, full-length 16S rRNA gene sequences (>1,400 bp in length) were extracted from representative MAGs with Barrnap v0.9 (Seeman, 0000) and placed into the SILVA 138 SSU NR99 reference phylogenetic tree (Quast et al., 2013; Yilmaz et al., 2014) using the SINA aligner (Pruesse, Peplies & Glöckner, 2012) and Maximum Parsimony algorithm of the ARB program (Ludwig et al., 2004). In total, 73 out of 136 species-representative MAGs had a complete 16S rRNA gene (>1,400 bp in length) and 23 of those, had more than one gene. The taxonomic assignment of sequences inferred from the SILVA database was used to replace the alpha-numeric taxonomic group names provided by GTDB where possible.

A phylogenetic tree was constructed using a concatenated alignment of 16 ribosomal proteins (L2, L3, L4, L5, L6, L14, L16, L18, L22, L24, S3, S8, S10, S17, S19) following a similar procedure to Hug et al. (2016). To represent the diversity as accurately as possible, the dataset was supplement with genomes of Bacteria and Archaea that were labelled as ‘Representative’ and ‘Complete’ in the RefSeq database (>2,500 genomes). Prodigal was used to predict coding sequences and target proteins were identified using hmmsearch v3.3.1 (Eddy, 2011) against PFAM HMM models for each ribosomal protein (E-value threshold of 1E -5). Individual gene sets were aligned using Muscle v3.8.15 (Edgar, 2004) (parameters: -maxiters 16) and trimmed using TrimAI v1.4.1 (Capella-Gutiérrez, Silla-Martínez & Gabaldón, 2009) (parameters: -automated1). All alignments were concatenated to form a single 16 gene alignment and a phylogenetic tree constructed using FastTree v.2.1.10 (Price, Dehal & Arkin, 2010) (parameters: -gamma –lg). The tree was visualised and annotated using iToL v4 (Letunic & Bork, 2019) (Fig. 3).

Figure 3 Phylogenetic diversity of metagenome assembled genomes (MAGs) from the Fram Strait.

The maximum likelihood tree was constructed from the concatenated alignments of 16 ribosomal proteins present in the MAGs and reference genomes of Bacteria and Archaea available in RefSeq. Coloured outer rings indicate groups that are represented by MAGs whilst circles on midpoints of the same colour indicate the exact position of MAGs within those groups.

Recovery of full-length 16S rRNA gene sequences

The extraction of 16S rRNA gene sequences from metagenomic reads can provide an insight into the community sampled and aid in identifying major taxonomic groups that are missed in the recovery of MAGs. Additionally in this study, it also allows for the comparison of the two sequencing methods. However, a major restriction with short-read metagenomic sequencing is the limited capacity to accurately reassemble full length 16S rRNA genes. With the advent of highly accurate long read sequences generated from PacBio sequel II (>99% accuracy), full length 16S rRNA genes can be retrieved from single reads without a need for assembly, thus circumventing previous limitations. This not only results in MAGs with complete 16S rRNA operons but also provides many additional sequences that can help to expand current databases without the need for a second, targeted sequencing run or additional sequencing platform. To provide future users of the data with a more detailed insight into the community and to demonstrate the value of long PacBio reads, we used the tool Barrnap v0.9 with default settings (parameters: –kingdom bac) to extract 16S rRNA gene sequences from the PacBio reads. A length cut-off was applied at 1000 bp, to focus on complete or near-complete gene sequences. For comparison, 16S rRNA reads were identified in the Illumina dataset using SortMeRna (Kopylova, Noé & Touzet, 2012) with a length cut-off of 120 bp. The extracted reads from each dataset were clustered into operational taxonomic units (OTUs) using the program CD-HIT-EST (Fu et al., 2012) at a 99% threshold. Reads were aligned using the SINA aligner and phylogenetically placed into the SILVA SSU_RefNR99 138.1 reference tree using the Parsimony tool in ARB, as described in ‘Phylogenetic assessment of MAGs’. The raw read numbers for the identified community were Hellinger transformed, compared using a Bray–Curtis dissimilarity matrix and visualised in a dendrogram format and ordinated using non-metric multi-dimensional scaling analysis (NMDS; Fig. S2) using the vegan package (Oksanen et al., 2013) in RStudio v1.1.463 (R Core Team, 2015). The taxonomic diversity of each sample was visualised using the ggplot2 package (Wickham, 2016) in RStudio v1.1.463 (Fig. 4) and the relative abundance of all taxonomic groups across samples is provided in Tables S4 and S5 (Illumina-derived 16S rRNA gene composition) and Table S4 (PacBio-derived 16S rRNA gene composition).

Figure 4 Phylogenetic composition of each metagenome sample derived from 16S rRNA gene sequences.

Genes were recovered from raw metagenomic reads using Barrnap (PacBio reads) and SortMeRNA (Illumina reads) and included only if the length was >1,000 bp or >120 bp respectively. Sequences from each dataset were clustered at a 99% identity threshold and taxonomically classified by inclusion into a reference phylogenetic tree from the SILVA 138 NR99 database. For clarity, the lettering used at the end of the sample name shows which PacBio and Illumina metagenomes came from the same sample, e.g., 4571_H and 4502_H.

Results

Identification of distinct water masses

The samples were identified as being of either Atlantic, Arctic or a mixed water mass origin based upon the measured abiotic parameters (Table S1). Those with a temperature < 0 °C and a salinity of < 34 psu were labelled as Arctic whereas those with a temperature of >5 °C and a salinity of ∼35 psu were labelled as Atlantic, in accordance with previous studies (Rudels et al., 2013; Fadeev et al., 2018). Samples with values in between these thresholds were defined as being of mixed origin.

Coverage and community

To determine the quality of the 14 metagenomes with respect to capturing the sampled community, NonPareil was used to calculate coverage; for this analysis, the long PacBio reads were sheared into 150 bp fragments to allow a direct comparison to the short Illumina reads. The coverage values across all metagenomes ranged from 0.72–0.88, indicating a high coverage regardless of sequencing method.

Prior to assembling the metagenomic reads, the community composition of each sample was assessed using the 16S rRNA gene (Fig. 4). Such an analysis provides an insight into the community sampled for future users of the data, whilst allowing comparisons between the two sequencing methods and between the sampled community and the phylogenetic diversity of MAGs recovered. In total, >950,000 16S rRNA gene fragments were recovered from short-read metagenomes along with >50,000 from long-read metagenomes. Within those recovered from long-reads, >16,000 were identified as full-length, highlighting the value of PacBio Sequel II sequences. Comparing the community composition identified across samples resulted in three distinct clusters reflecting the Artic, Atlantic and mixed water masses (Fig. S2) that were identified based on variations in abiotic parameters (Table S1).

Assembly

The eleven short-read metagenomes, containing between 46–81 million reads post quality trimming, were individually assembled using Megahit, generating assemblies with an average total length of 1.05 Gbp, N50 value of 691 and 2.1 million contigs (Table 1). The three long-read metagenomes were individually assembled using metaFlye which resulted, on average, in shorter assemblies than the short-read metagenomes, 374 Mbp in length, but with much higher N50 values, 67 kbp, and a lower number of contigs, 9063 (Table 1). The average longest contig length was also significantly larger in the long-read assemblies, 1.99 Mbp, compared to the short-read assemblies, 0.34 Mbp.

Binning

The contigs longer than 2.5 kbp were used to recruit reads from all short-read Illumina metagenomes using BBmap, to provide as much coverage variation information as possible to the binning tools. After binning with three different tools, DasTool recovered a consensus set of 349 bins that were >50% complete and < 10% contaminated. All bins were manually refined using anvi’o before being dereplicated into 136 species-level clusters, based on a 95% ANI threshold (Table S2). The selected species-representative MAGs had completeness values between 52–100% and between 0–9% contamination with an average genome size of 1.94 Mbp (Table S2). Of these, 27 species-representative MAGs were classified as high-quality drafts according to the MIMAGs standards(Bowers et al., 2017); meaning they contain a 23S, 16S and 5S rRNA gene and at least 18 tRNAs with a completeness value of >90% and contamination < 5%. Furthermore, owing to the long PacBio reads, 75 of the MAGs had a 16S rRNA gene present and 35 of the MAGs were composed of < 10 contigs, indicating a very high contiguity.

The phylogenetic classification of MAGs was performed using two different approaches to ensure robustness and reliability, these included a 16S rRNA gene approach, where the genes were present, and a single copy marker gene and phylogenetic approach through the GTDBtk tool. The diversity captured by species-representative MAGs was then visualised through the reconstruction of a phylogenetic tree using a concatenated alignment of 16 ribosomal proteins from the MAGs and >2500 genomes labelled as ‘Complete’ and ‘Representative’ in the RefSeq database (Fig. 3). The recovered diversity encompassed 9 phyla, 11 classes, 27 orders, ∼51 families and ∼54 genera. The most species-rich taxa were the Flavobacteriales (41 species), Pseudomonadales (18 species) and Rhodobacterales (17 species). This picture of community diversity obtained from the MAGs is comparable to that from the 16S rRNA gene sequences alone, even though we are able to recover a much higher number of 16S rRNA gene sequences.

Relative abundance of MAGs

The range in estimated relative abundance values across the MAGs were from < 0.001–10.75%, with the lower values being attributed to the recruitment of reads to MAGs from compositionally different metagenomes (Tables S4 and S5). Summing the relative abundance values in each sample indicated that the species-representative MAGs accounted for 52.3–89.7% of the community, further supporting that the recovered MAGs covered all of the major taxonomic groups present.

Discussion

The Fram Strait is not only a region of global significance due to its role in heat and water mass exchange but also as it provides an invaluable opportunity to study ecological changes from the Atlantic to Arctic Ocean. Although this region has been studied extensively in recent years, there is still only limited information available on the ecology of microbial communities. The metagenomics and MAG dataset presented here is derived from samples collected across the Fram Strait region and provides unique genetic resources represented in contrasting water masses of Arctic, Atlantic and mixed origin. The dataset also provides a valuable combination of short-read and long-read metagenomes, representing one of the first PacBio Sequel II metagenome and MAG dataset from marine environmental samples.

The distinct water masses sampled across the Fram Strait are distinguishable based on temperature and salinity (Rudels et al., 2013) and are shown here to harbor unique microbial community compositions (Fig. 4 and Fig. S2). One major distinction is the elevated proportions of Flavobacteria taxa (such as Aurantivirga, Formosa and NS5) in the Atlantic (West Spitsbergen Current; WSC) compared to the Arctic (East Greenland Current; EGC) water mass, which is likely influenced by the time of sampling (July–August). Seasonal phytoplankton blooms in the WSC region have been well evidenced and shown to reach maximum integrated chlorophyll a values of 100 mg/m3 (Nöthig et al., 2015). Summer phytoplankton blooms typically occur from June to July (Nöthig et al., 2015) and lead to the enrichment of Flavobacteria, with intermittent peaks of specific taxa (Formosa, Polaribacter and NS5) (Wietz et al., 2021) resembling successional patterns that are known from temperate spring phytoplankton blooms (Teeling et al., 2012). In comparison, the EGC does not experience such pronounced phytoplankton blooms and instead it has been suggested that a different food web-based structure may exist in these waters (Wietz et al., 2021). In agreement with previous findings (Fadeev et al., 2018), the EGC was enriched in Gammaproteobacteria (SAR86, SUP05) and taxa related to Arctic winter and deeper waters (Marinimicrobia and SAR324). Between these distinct water masses, the central Fram Strait region is subject to complex and dynamic hydrographic processes with lateral mixing, advection of Atlantic water under Arctic water and westward flowing-mesoscale eddies originating from the WSC all exerting an influence over different spatial and temporal scales. Due to such complexities in determining these features, we defined the samples from the central Fram Strait region, whose abiotic parameters were between the thresholds of Arctic and Atlantic water, as ‘mixed’. These mixed origin samples were shown to harbor the highest proportion of taxa within the Cryomorphaceae and Flavobacteriaceae (such as Polaribacter and NS9) as well as consist of up to 10% Verrucomicrobia that was in low (<2%) abundance in the other water masses. Recently, an investigation into a mesoscale filament in the central Fram Strait region revealed an increase in phytoplankton productivity, microbial cell counts and specific taxa related to phytoplankton-derived organic matter degradation (Fadeev et al., 2021). Although in the data presented here, the measured fluorescence values (Table S2) were not indicative of a bloom event, the enriched taxa are known as key players in organic matter degradation. Therefore, it is possible that the mixed samples are derived from a mesoscale filament or eddy and represent a post-phytoplankton bloom situation.

Integrating short and long read sequencing technologies to recover microbial populations

The pipeline we employed to process metagenomics reads was carefully optimized to ensure high quality and accurate assemblies and to maximize the number of near-complete MAGs recovered. The assembly of reads for each metagenome individually as opposed to using a co-assembly approach, likely reduces the chance of chimera formation and prevents the loss of strain variation across populations. The subsequent binning of contigs, after removing those less than 2.5 kbp in length to minimize misbinning (Chen et al., 2020), was performed using multiple tools as opposed to a single tool as this approach has recently been shown to greatly increase the number of reconstructed near-complete genomes (Probst et al., 2017; Sieber et al., 2018). To further improve the completeness and contiguity of bins, we reassembled reads that were recruited to bins of the same taxonomic class within each sample. Although the resulting set of bins were of seemingly high quality, it is well known that using automated tools can result in misbinning of contigs due to similarities in sequence composition and coverage across genomic regions of different microbial populations (Chen et al., 2020). Therefore, each of the generated bins was visually inspected and subject to manual refinement which involved the removal of misplaced contigs and the discarding of erroneous bins.

By employing long-read and short-read sequencing, we are able to compare the number and quality of MAGs retrieved between both platforms. Of the species-representative MAGs recovered, those generated from the PacBio metagenomes had, on average, larger genome sizes, higher N50 values and were less fragmented compared to those retrieved from Illumina metagenomes (Table S2). One of the major limitations of short-read metagenomics is the low recovery rate of rRNA genes within MAGs. However, in this study, 84% of MAGs retrieved from the PacBio metagenomes contained at least one complete 16S rRNA gene sequence, highlighting another key advantage of using long Hifi reads. Therefore, we can conclude that HiFi read metagenomes derived from the PacBio Sequel II platform can greatly improve the number and quality of MAGs recovered, which will allow for further advancement in our understanding of the ecology of marine microbial communities.

Conclusion

The aim of this manuscript was to provide a useful data resource to supplement future ecological analyses on Arctic microbial communities and to help alleviate biases against metagenomic sequence data from polar regions. The generation of 14 metagenomes from short and long read sequencing platforms along with 136 manually-refined species-representative MAGs provides a valuable dataset to address questions regarding distribution of taxa and functionality on a community- and species-level as well for downstream comparative genomics.

An initial insight into the composition of the metagenomes using 16S rRNA gene sequences revealed taxonomically-rich communities with distinct compositions corresponding to the different water masses sampled. The recovery of more than 16,000 full-length 16S rRNA gene sequences from raw PacBio reads can allow for further high-resolution phylogenetic analyses to be performed. The diversity captured by the 136 manually-refined species-representative MAGs encompassed more than 50 genera and consisted of members from all major taxonomic groups in the sampled community. Furthermore, the majority of MAGs recovered were of high quality, with 27 MAGs being classified as high-quality drafts according to MIMAGS standards, 75 MAGs containing at least one 16S rRNA gene and 35 MAGs having < 10 contigs.

The pipeline used to process the metagenome data and recover the described MAGs was thoroughly tested and optimized at each stage to ensure reliable and high-quality results. Although the provided data is suitable for direct inclusion in further analyses, it is recommended to confirm any of the stated values here using the most up to date analysis tools, particularly with respect to MAG completeness, contamination and taxonomic classification.

Data records

All data provided in this study has been deposited in the European Nucleotide Archive (ENA) at EMBL-EBI under accession number PRJEB41592.

Supplemental Information

Supplemental Information 1 Supplementary Tables

Click here for additional data file.

Supplemental Information 2 Visualisation of sequence over length

Upper four plots are example outputs of FastQC that summarises the per base sequence quality of Illumina reads. The bottom two plots are example outputs of NanoPlot that summarises the average read quality for different read lengths.

Click here for additional data file.

Supplemental Information 3 Comparison of metagenomic samples based on Bray–Curtis dissimilarity of 16S rRNA gene composition

(A) Dendogram generated from Bray–Curtis dissimilarity matrix of samples’ community composition at a genus level, (B) Non-metric multi-dimensional scaling ordination of Bray–Curtis dissimilarity of samples’ community composition at a genus level.

Click here for additional data file.

We would like to thank Jörg Wulf and Mirja Meiners, from the Molecular Ecology department at the Max Planck Institute for Marine Microbiology in Bremen, for their technical support. Furthermore we would like to thank the team at the Max Planck Genome Centre in Cologne for their efforts with sequencing the samples.

Additional Information and Declarations

Competing Interests

Author Contributions

Data Availability

The authors declare there are no competing interests.

Taylor Priest conceived and designed the experiments, performed the experiments, analyzed the data, prepared figures and/or tables, authored or reviewed drafts of the paper, and approved the final draft.

Luis H. Orellana, Bernhard M. Fuchs and Rudolf Amann conceived and designed the experiments, authored or reviewed drafts of the paper, and approved the final draft.

Bruno Huettel performed the experiments, authored or reviewed drafts of the paper, and approved the final draft.

The following information was supplied regarding data availability:

All data in this study are available in the European Nucleotide Archive (ENA) at EMBL-EBI: PRJEB41592.

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
