# Peer review of "Microbial metagenome-assembled genomes of the Fram Strait from short and long read sequencing platforms"

_PeerJ, doi:10.7717/peerj.11721_

## Round 0.1 · original submission · Minor Revisions

Thanks for submitting your work to PeerJ. You will see that the reviewers have suggested several areas of clarification. Reviewer 1 has asked that you expand your discussion to include additional details regarding the genomic features of the microbial communities in this region. Specifically, the reviewer would like to see the inclusion of additional literature, highlighting the region studied. Reviewer 2 has asked for clarification regarding some of the terms used in the study (e.g. mixed origin) and some clarifications on the figure captions.

·

Basic reporting

Priest et al., present an interesting research paper from the Fram Strait, a body of water that links the Arctic and North Atlantic Oceans. Here, the authors generated short- and long-read metagenomes to reconstitute metagenome-assembled genomes (MAGs) with a view to describe the microbial functionality in this important region. The paper was easy to follow and the conclusions generally satisfactory.
My major concern is that the results and discussion and very much focused on the MAGs generated from a technical perspective. Although the technical quality of this work is very high, there is nothing that reminds us of the value of the Fram Strait by the end of the paper. Since this region is important I expected to see analyses of interesting genes or the like. Alternatively, some literature that describes how unique this region is taxonomically would be valuable also. Only Discussion paragraph 1 (Lines 276-287) mentions the system, but in fairly broad terms.

Experimental design

The experimental design is certainly the strength of the paper. I believe that the techniques applied are well explained and justified.
Since the authors also checked for 5S and 23S rRNA genes, as well as at least 18 tRNAs, please add this information to Table S2.

Validity of the findings

In my opinion the findings are valid and expand what is known about the Fram Strait in terms of microbial genomes. Again, I do wish that the findings touched on the genomic features of these microbes to explain adaptations to this interesting region.
Finally, since this work is among the first to contrast short- and long-read sequencing and assembly to generate MAGs, I would like to see which insights were provided because of the long-read sequencing. In other words, what did the short-reads miss that was captured by long-reads. This would potentially assist readers in opting for long-read sequencing in the future. For example, reconstructing MAGs easier or accessing full-length 16S rRNA genes from MAGs.

Additional comments

The final lines of the abstract (L33-35) are unnecessary in my opinion. This is more of a technical note and doesn't inspire the reader.
Dendrogram is spelt "dendogram" in main text (L204) and supplementary file 2.
Line 137 "*.sam files" should be "sequence alignment maps"
Line 207: "CTD" I'm aware that oceanic researchers know this term, but it may be unclear to a general audience.
Could the nonpareil curves be a supplementary figure?
The table names in the supplementary table don't correspond between the tab name and the table name on the sheet. This makes it confusing to follow.

Reviewer 2 ·

Basic reporting

The manuscript is well written, clear and easy to follow. No obvious spelling or grammatical errors were detected. The article structure is acceptable and the raw data accessible.

Please correct:

Line 114-115: This should be reported in the Results section. Please remove as it is already present in the Results section.
line 209: "phylogenetic diversity". This should rather be "taxonomic diversity" or "taxonomic composition". The same applies to the legend for Figure 4.

Experimental design

The experimental design was based on the collection, sequencing, assembly and reporting of MAGs from the Fram Strait. This included both short and long reads sequencing technologies. This was done to supplement the number of metagenomic samples/data sets currently available for this region which is generally low. Numerous analysis protocols were implemented and a sound methodology followed with regards to analysis and investigation. Albeit relatively sparse with regards to sample numbers, the inclusion of a long read data does add some weight to the manuscript. The methods are adequately described and reproducible.

Please add:

In Figure 1 and Table 1 please clearly identify the origin of the sample, i. e. "Atlantic", "Arctic", "Mixed"

Validity of the findings

As stated above, the number of samples are relatively low but the data is strengthened by the inclusion of the PacBio data. The impact and novelty of the findings are low but as indicated these are not assessed. The authors incorporated various currently accepted applications and tools to produce MAGs which was described as the main objective of the paper. All the data has been provided and is accessible.

Areas of concern:
"Mixed" origin samples - In line 282 it is stated that "likely of mixed origin" yet in line 230 it is stated "mixed water masses sampled". The authors should clearly identify if this is speculation or if the samples were obtained from a "mixed" source. Did the authors use "Mixed" because these samples did not cluster with the other groupings?

Please address this comment clearly as it has implications for the future use of these datasets by other researcher which is the underlying basis of this publication.

Furthermore,

Line 283: "changes in temperature and salinity". These cannot be inspected as the location, i. e. "Arctic", "Atlantic", "Mixed", is not included in S1.
Line 285: "observable differences". How was this concluded? Visually?
Line 312: "distinct patterns in taxa distribution". Please expand, give examples and indicate how these "distinct patterns" were inferred. Is this only based on the clustering? If yes, we return to the comment above about the "Mixed" samples.

Additional comments

The manuscript has details what the authors set out to do, which is the production of MAGs from the Fram Strait region.

That being said, it is a pity that not more was done with the data. It does seem that energy was lost as so much more is possible.

The methodology and analysis is sound and the availability of the raw and assembled datasets will assist in future research. The addition of long read sequencing data does give more weight to the publication and the pipeline described in the analysis of the long read data may be of importance to other researchers.

As stated above, please clarify and clearly label the sample origins

---

## Round 0.2 · accepted · Accept

Thanks for submitting your work to PeerJ. I am happy with your responses to the reviewer comments. Your work is well written and will represent an excellent resource to the community.